# Fetal MRI Analysis of Corpus Callosal Abnormalities: Classification, and Associated Anomalies

**DOI:** 10.3390/diagnostics14040430

**Published:** 2024-02-15

**Authors:** Kranthi K. Marathu, Farzan Vahedifard, Mehmet Kocak, Xuchu Liu, Jubril O. Adepoju, Rakhee M. Bowker, Mark Supanich, Rosario M. Cosme-Cruz, Sharon Byrd

**Affiliations:** 1Department of Diagnostic Radiology and Nuclear Medicine, Rush Medical College, Chicago, IL 60612, USA; farzan_vahedifard@rush.edu (F.V.); mehmet_kocak@rush.edu (M.K.); xuchu_liu@rush.edu (X.L.); jubril_o_adepoju@rush.edu (J.O.A.); sharon_byrd@rush.edu (S.B.); 2Department of Pediatrics, Division of Neonatology, Rush Medical College, Chicago, IL 60612, USA; rakhee_bowker@rush.edu; 3Department of Radiology and Nuclear Medicine, Division for Diagnostic Medical Physics, Rush University Medical Center, Chicago, IL 60612, USA; mark_supanich@rush.edu; 4Department of Psychiatry and Behavioral Sciences, Rush Medical College, Chicago, IL 60612, USA; rosario_m_cosme@rush.edu

**Keywords:** fetal MRI, corpus callosal abnormalities, central nervous system anomalies, weeks of gestation, probst bundles

## Abstract

Background. Corpus callosal abnormalities (CCA) are midline developmental brain malformations and are usually associated with a wide spectrum of other neurological and non-neurological abnormalities. The study aims to highlight the diagnostic role of fetal MRI to characterize heterogeneous corpus callosal abnormalities using the latest classification system. It also helps to identify associated anomalies, which have prognostic implications for the postnatal outcome. Methods. In this study, retrospective data from antenatal women who underwent fetal MRI between January 2014 and July 2023 at Rush University Medical Center were evaluated for CCA and classified based on structural morphology. Patients were further assessed for associated neurological and non-neurological anomalies. Results. The most frequent class of CCA was complete agenesis (79.1%), followed by hypoplasia (12.5%), dysplasia (4.2%), and hypoplasia with dysplasia (4.2%). Among them, 17% had isolated CCA, while the majority (83%) had complex forms of CCA associated with other CNS and non-CNS anomalies. Out of the complex CCA cases, 58% were associated with other CNS anomalies, while 8% were associated with non-CNS anomalies. 17% of cases had both. Conclusion. The use of fetal MRI is valuable in the classification of abnormalities of the corpus callosum after the confirmation of a suspected diagnosis on prenatal ultrasound. This technique is an invaluable method for distinguishing between isolated and complex forms of CCA, especially in cases of apparent isolated CCA. The use of diffusion-weighted imaging or diffusion tensor imaging in fetal neuroimaging is expected to provide further insights into white matter abnormalities in fetuses diagnosed with CCA in the future.

## 1. Introduction

The corpus callosum (CC) serves as the primary cerebral commissure in the supratentorial region of the brain [1]. There are two smaller interhemispheric fissure connections next to the CC, namely, the anterior commissure and hippocampal commissure. The corpus callosum primarily consists of white matter fibers connecting the two cerebral hemispheres at the midline. There are more than 200–250 million myelinated axons crossing the midline in the corpus callosum during brain development, establishing crucial connections between the hemispheres [2].

During the sixth gestational week (GW), the commissural plate begins to differentiate, followed by the crossing of pioneer axons [3]. As corpus callosum forms between the eighth and fourteenth weeks of gestation, callosal precursors and cortical fibers develop in bilateral cerebral hemispheres. Anatomically, it consists of four components: the rostrum, the genus, the body, and the splenium. The genus is the first part of the corpus callosum to exhibit crossed fibers, followed by the body and the splenium, and the rostrum was considered the last part of the corpus callosum to show crossed fibers [4]. However, advanced techniques like diffusion tensor neuroimaging (DTI) could identify the rostrum at 15 weeks of gestational age, along with the anterior part of the body and the genus. The posterior part of the body and splenium develop in the later part of the gestation and complete by 18 to 19 gestational weeks, highlighting the importance of gestational age in describing the corpus callosal abnormalities [5,6,7].

An intricate and tightly regulated sequence of developmental processes culminates in the corpus callosum during gestation and into adulthood [8]. If any of these events are disrupted, it can result in corpus callosal abnormalities (CCA), which is a rare condition. In accordance with data from the National Organization for Disorders of the Corpus Callosum, a person in every 2053 experiences symptoms associated with abnormalities in the corpus callosum. Its frequency is between 0.5 and 70 per 10,000, and its prevalence among children with developmental disorders is about 230 to 600 per 10,000 (2.3%) [9,10,11,12]. It is found in one in every 19,000 autopsies [13]. The incidence is higher among boys than girls [14]. Corpus callosum (CC) abnormalities can occur alone; however, they are more commonly associated with a wide range of other abnormalities of the central nervous system or malformations of other organ systems [15].

Heterogeneous terminologies are used in the classification of corpus callosal abnormalities. Different classification systems are based on either anatomical or functional bases. Barkovich et al. previously divided CCA into dysgenesis (a defective development) and hypogenesis (an incomplete formation) [16]. Another classification by Santo et al. subdivided CCA into complete agenesis and partial agenesis, grouping hypogenesis and dysgenesis into partial agenesis [17]. They further classified them into isolated versus complex CCA. Witelson et al. used DTI and classified CCA on a functional basis [18]. Al Hashim et al. used an embryology-based functional anatomical classification system and categorized them into complete CCA, anterior CCA, posterior CCA, and complete hypoplasia [19]. Hanna et al. updated the previous classification of CCA by subdividing them into complete agenesis, hypoplasia without dysplasia, hypoplasia with dysplasia, and dysplasia. The term partial agenesis was not used in this classification [20]. A more accurate molecular classification of CCA patients will be possible with the ability to evaluate newly identified genes based on similarities in the corpus callosum appearance [20]. This classification provides valuable information for future studies regarding the role of genetic mechanisms in CCA [20,21]. 

Complete agenesis has two types. In type 1, axons are present, but the commissural fibers that develop by crossing the midline are absent. Probst bundles are those uncrossed fibers that run along the superior and medial areas of the bilateral lateral ventricles [22,23]. In type II, axons are absent, and thus, Probst bundles are not seen [24]. Hypoplasia refers to a uniformly thin CC or underdeveloped posterior corpus callosum with intact morphology. In callosal hypoplasia, sigmoid bundles represent a heterotopic connection between one hemisphere’s frontal lobe and the other hemisphere’s occipital lobe [25]. Dysplasia refers to the defective development of CC, which has an abnormal shape [26]. 

As development starts anteriorly and progresses posteriorly, the earlier the callosal development is arrested, the smaller the genus and anterior body are formed [27]. In contrast, with the sequential arrest of growth of the corpus callosum in congenital hypogenesis, acquired defects can affect any of the parts of the corpus callosum [27]. A number of reasons are involved in causing insults to the developing brain, including infectious diseases (Toxoplasmosis, Rubella, Cytomegalovirus, and Herpes), toxic/metabolic substances (like alcohol), chromosomal/genetic factors, and vascular disorders [28]. Many syndromes are associated sporadically with CCA, like Aicardi syndrome, Shappiro syndrome, Anderman syndrome, and Acrocallosal syndromes [29,30,31,32,33,34].

The diagnosis of CCA by fetal ultrasound has a high false-positive rate, and its range varies from 0% to 20% depending on the operator [17]. Fetal magnetic resonance imaging (MRI) for evaluating fetal CC is considered superior because of its multiplanar capabilities and spatial resolution [26]. A fetal MRI not only confirms CCA but also determines its structural morphology more accurately while detecting coexisting anomalies like gyration anomalies and heterotopia not visualized on prenatal ultrasound [17]. A systematic review of fetal MRI detected additional abnormalities in 22.5% of cases compared to ultrasonography in prenatal cases [35]. Poor postnatal neurodevelopmental outcome is even more common to occur in fetuses with CCA associated with other CNS anomalies [36]; hence, MRI should be performed as part of CCA assessment, particularly in fetuses with apparently isolated CCA. With the advent of ultra-fast T2 imaging techniques, fetal MRI allows direct visualization of CCA on a midline sagittal image even without volume imaging or multiplanar reconstruction.

In conclusion, understanding the intricate development and potential callosal abnormalities is essential for unraveling the complexities of cerebral connectivity. With various classification systems shedding light on the diverse manifestations of corpus callosal abnormalities (CCA) and advanced imaging techniques like fetal MRI enhancing diagnostic precision, our study endeavors to contribute to this knowledge landscape. By adopting the refined classification system proposed by Hanna et al., we aim to categorize 24 individuals with callosal anomalies on fetal MRI, and unravel the nuances of these variations, and their associated neurological and non-neurological abnormalities [20]. To date, very few studies have used this latest classification to characterize neuroimaging findings in fetuses with CCA.

## 2. Methods

### 2.1. Study Population and Design

In this study, retrospective data from antenatal women who underwent fetal MRI between January 2014 and July 2023 at Rush University Medical Center was analyzed for CCA. The fetal maternal unit referred all these cases to the diagnostic neuroradiology department for fetal MRI upon detecting corpus callosal anomalies on prenatal ultrasound. The study received approval from the Institutional Review Board, and explicit consent was waived for the study. The study included fetuses with abnormalities of the corpus callosum detected on fetal MRI and suspected related congenital anomalies that were not adequately assessed on ultrasound. A number of participants were excluded from the study, including those with MRI contraindications (pacemaker, metal foreign body, or aneurysm clip), poor image quality, those without the abnormal corpus callosum of the MRI, those with secondary causes of callosal abnormalities (examples include congenital hydrocephalus, Chiari malformations, intracranial hemorrhage, metabolic disorders, infection, or toxins), those without genetic analysis, and those with disagreement among neuroradiologists about CCA diagnosis. A total of 140 fetal MRIs were analyzed for CCA; 30 fetal MRI’s had abnormal corpus callosum, but six cases were not included because they met exclusion criteria. In total, 24 antenatal women participated in the study.

### 2.2. Data Collection and Management

Chart reviews were performed, and the following data were extracted from the medical records: maternal age at delivery, maternal comorbid diagnoses, number of gestations, maternal race/ethnicity, mean weeks of gestation (GW) while undergoing fetal MRI, route of delivery, known teratogenic exposures during pregnancy, amniocentesis results if performed, birth GA (completed gestational weeks), sex, and length of stay (LOS) in the NICU (Neonatal intensive care unit). The pregnancy outcomes were categorized as termination of pregnancy, spontaneous intrauterine fetal demise, inborn live birth, or death in the neonatal period. In this study, clinical information was obtained from the pediatric neurology and genetics clinic. All the antenatal women in our study had genetic evaluation with karyotyping, chromosomal microarray analyses, and whole exome sequencing (WES).

### 2.3. Fetal Magnetic Resonance Imaging

At the study institute, fetal MR scans were conducted using an Espree 1.5-T MR scanner manufactured by Siemens Medical Solutions in Erlangen, Germany. The scans were performed without any sedation. Imaging was performed using a HASTE pulse sequence. The parameters include: Time to echo (TE): 120 ms; repetition time (TR): 4300 milliseconds (ms); field of view (FOV): 23 (phase) × 23 (frequency) cm; slice thickness (ST): 3 mm; interslice spacing (IS): zero; matrix: 256 (phase) × 180 (frequency) in axial, sagittal, and coronal projections. This T2-weighted MR sequence uses half-fourier single-shot turbine spin echo (abbreviated HASTE) to acquire images in one second.

To ensure an unbiased evaluation, two independent board-certified neuroradiologists (S.E.B. and M.K. with over 15 years’ experience) reviewed the fetal MR brain images of participants without knowledge of their clinical histories. All cases were categorized according to the proposed classification system. Throughout the study, all authors remained blinded to any identifiable information, and strict confidentiality was maintained regarding protected health information. The final diagnosis was arrived at by their consensus. Neuroradiologists could establish concordance in 21 cases (92%). The non-concordance was noted in the diagnosis of one case in each class of hypoplasia with dysplasia, dysplasia, and complete agenesis. The interobserver agreement statistics showed kappa statistics between 0.86 and 1 for the presence of abnormal callosum, and between 0.79 and 1 for the presence of other brain anomalies.

### 2.4. Classification of CCA on Fetal MRI

Patients with CCA are further subclassified into complete agenesis, hypoplasia without dysplasia, dysplasia, and hypoplasia with dysplasia [20]. Complete agenesis of the corpus callosum was defined as the absence of the corpus callosum in the mid-sagittal plane. The term hypoplasia refers to a uniformly thin or underdeveloped posterior part of the corpus callosum with intact morphology. Dysplasia refers to defective CC development, which has an abnormal shape (Figure 1). The normative indices for the corpus callosal length for fetuses between 18–22 weeks were derived from the study by Parazzini et al. [37], while between 22–37 weeks were derived from the study by Garel et al. [38]. A mean corpus callosum size of 18.8 to 42.6 mm was reported across gestational ages 19–36 weeks in the study conducted by Harreld et al. [39]. In our 2D MR images, the voxel size is 1.3 × 0.9 × 3.0 mm in the X, Y, and Z directions. The size of the corpus callosum fits within the dimensions of the voxel. The corpus callosum length and width fall within the specified voxel size dimensions (1.3 mm and 0.9 mm). The corpus callosum thickness is within the Z (3.0 mm) dimension of the voxel size. Thus, our imaging resolution is sufficient to capture details of the corpus callosum in all three dimensions, supporting the reliability of our fetal MRI measurements.

The presence of ventriculomegaly, colpocephaly, Probst bundles, and absent septum pellucidum were considered secondary to abnormal corpus callosum rather than associated CNS anomalies. Fetal ventriculomegaly is subclassified into mild form (10–12 mm in transverse diameter), moderate form (12–15 mm), and severe form (>15 mm) [40]. We further classified CCA into associated CNS anomalies, and non-CNS anomalies. We reviewed fetal brain MRI images for the following associated CNS abnormalities: hypoplastic or dysplastic brainstem, Dandy-Walker complex and other posterior fossa anomalies, hydrocephalus, interhemispheric cysts, optic nerve anomalies, septal anomalies, and migrational anomalies. Non-central nervous system abnormalities were reviewed separately by a pediatric body radiologist. The anterior commissure and hippocampal commissure are not studied for the classification system used in this study.

### 2.5. Statistical Analysis

Data were obtained and entered into Microsoft Excel LTSC MSO (v.2401 Build 16.0.17231.20194) 64-bit. Parametric tests were conducted because the data were normally distributed. Means and their corresponding standard deviations are analyzed for the quantitative variables. Frequencies were used as percentages to represent the qualitative variables.

## 3. Results

24 fetuses with abnormal corpus callosum on fetal MRI were recruited for final analyses. Twenty-three of the cases involved singleton pregnancies, and one involved a twin pregnancy. Antenatal women in the study range in age between 16 and 41 years, with a mean age of 30 years (standard deviation ±7). The gestational age of the fetuses was between 19 and 36 weeks, with a mean age of 26 WG (standard deviation ±5 days). In one case, the parents were consanguineous, and in all other cases, it was sporadic. There were 55% (13/24) male fetuses and 45% (11/24) female fetuses among the participants. Genetic abnormalities were identified in seven (28%) cases, including trisomy 8 (12%), trisomy 18 (4%), 14q pathogenic deletion (4%), 17q12 pathogenic deletion (4%), and 46XX/47XX, +small marker chromosome mosaicism (4%) (Figure 2). Among seven antenatal women with abnormal genetic analysis, three are teenagers, two are elderly (>35 years), and the other two are within the normal age range.

Two antenatal women (8%) terminated their pregnancies after fetal MRIs were performed, while most women (91.6%) continued their pregnancy until delivery. Of them, two had stillbirth, and the remaining 20 had live-born babies who were admitted to the neonatal intensive care unit. 12.6% (3/20) of liveborn infants admitted with CCA died before NICU discharge. The maternal and birth characteristics of the study population are described in Table 1.

We classified these 24 fetuses based on the type of corpus callosal structural abnormality found on the fetal MRI midline sagittal image. 19 patients had complete agenesis (79.1%), three patients had hypoplasia (12.5%), and one patient each had dysplasia (4.2%) and hypoplasia with dysplasia (4.2%), respectively (Figure 3).

Individuals within each subclass of CCA displayed a range of abnormalities, both CNS and non-CNS. Out of 24 cases, 4/24 of them (17%) had isolated CCA, and 20/24 (83%) cases had complex forms of CCA with associated CNS and/or non-CNS anomalies (Figure 4).

Colpocephaly is seen in 13/24 cases (54%) (Figure 5). Of them, 12 cases had complete agenesis, and one case had hypoplasia. Absent septum pellucidum is seen in 5/24 cases (20.8%) (Figure 6); 4 cases had complete agenesis, and one case had hypoplasia. Probst bundles are identified only in the isolated form of CCA and account for 3/19 cases (15.7%) of complete agenesis and 3/4 cases of the isolated form of CCA (Figure 7). Probst bundles are not seen in other subclasses of CCA. Ventriculomegaly is seen in 7/24 cases with an incidence rate of 0.29%, 5 cases had complete agenesis, and 1 case each had hypoplasia and hypoplasia with dysplasia.

Associated CNS anomalies include 9/20 cases of interhemispheric cysts seen in 8 cases of complete agenesis and 1 case of hypoplasia. Out of 9 cases of IHC, 2 IHC were communicating with lateral ventricles. 4/20 (20%) cases had migration abnormalities such as gray matter heterotopia (Figure 8) in 3 cases of complete agenesis and hemimegalencephaly with schizencephaly in a case of dysplasia. 2 cases of cerebral cysts (2/20), including an arachnoid cyst in a case of complete agenesis class and a germinolytic cyst in a case of dysplasia. 3/20 cases of posterior fossa cysts, likely Blake’s pouch cysts, 2 in the complete agenesis class and 1 in the hypoplasia class. 2 cases of vermian hypoplasia (2/18) were noted, one each in the complete agenesis and hypoplasia classes (Figure 9 and Figure 10). A case of complete agenesis class had cerebellar hypoplasia and brain stem hypoplasia with a normal vermis. A case of complete agenesis class had a neural tube defect in the form of a sphenoidal meningoencephalocele. A case of dysplasia had diffuse thinning of white matter. 

Regarding non-neurological anomalies in the complex form of CCA, 2 cases had non-CNS anomalies without other CNS anomalies; one case had a nasal dermoid cyst, while another case had T12 hemivertebra and thoracic body fusion. A total of 4 cases had both CNS and non-CNS anomalies; 1 of the cases had severe hydrocephalus, hemimegalencephaly, a germinolytic cyst, and facial dysmorphism secondary to fused eyes. The second case had heterotopia and hydroureteronephrosis, and another case had sphenoidal meningomyelocele and a cleft lip with a cleft palate. The last case had an interhemispheric cyst, an arachnoid cyst, and a thick nuchal fold called hydronephrosis (Table 2). 7/19 cases of complete agenesis, 1/3 cases of hypoplasia, and 1 case each of dysplasia and hypoplasia with dysplasia showed a delayed neurodevelopmental outcome.

## 4. Discussion

The study included 16 to 41-year-old pregnant women, with a mean age of 30 ± 7 years, similar to the study performed by Yeh et al. [41]. In this study of a population of 24 fetuses diagnosed with CCA, the male to female ratio is 1.2:1. Yeh et al. and Manganaro et al. found similar results for corpus callosal abnormalities in prenatally diagnosed fetuses [41,42]. Jeret et al. reported similar results in postnatally diagnosed fetuses [12]. The development of corpus callosum completes by around 18 to 19 WG when detected in prenatal imaging [28]. Therefore, the gestational age in our study ranged from 19 to 36 weeks, with a mean of 26 (standard deviation ±5 days), consistent with the study of Tang et al. [43]. Sporadic cases accounted for 23 of 24 cases, with only one case resulting from consanguinity, which is consistent with other studies on prenatally detected CCA [44]. Current evidence suggests that both acquired and genetic mechanisms play an important role in the etiology of CCA [42]. Roughly 30% of the study participants had identifiable diagnoses, including cases of single-gene disorders, mosaicism, copy number variants, and trisomy’s. Many genetic disorders have been linked to CCA, including trisomy’s, gene deletion syndromes, copy number variants, and X-linked disorders [3,19]. The genetic variants noted in the study have also been identified in many other studies [45,46,47,48,49]. The trisomy 8 mosaicism is the most common (13%) genetic abnormality detected in our study. We detected 4/24 cases (25%) of syndromic corpus callosal agenesis, of which 3 patients had trisomy 8 mosaicism and one patient had trisomy 18. The percentage of clinical syndromic diagnosis is similar to the study performed by Al-Hashim et al. (22%), Bedeshi et al. (33%), and Schell- Apasik et al. (32%) [19,36,50]. 2 cases had a single gene disorder with pathogenic gene deletion at 14q24.3-32 and 17q12. Till date, only a few cases have been reported with 14q24.3-32 gene deletion in cases with corpus callosum agenesis [47]. 17q12 pathogenic deletion was identified in a few studies analyzing genetic variants in corpus callosal agenesis [48,49]. A total of (29%) cases had identifiable genetic causes, similar to the study conducted by Al-Hashim et al. [19]. Regarding the disposition of pregnancy, 2 cases underwent the termination of pregnancy (abortion), 2 cases had still given birth, and most of them continued their pregnancy until the 37th WG. These findings can be comparable to the study conducted by Ibrahim et al. [44]

The diagnosis of corpus callosal abnormalities in the axial planes in fetal MRI allows suspicion of the pathology, while the mid-sagittal plane is necessary to make the diagnosis of CCA without the need for 3D reconstruction techniques or volumetric analysis. Nevertheless, volumetric segmentation should be considered, especially in cases with suspected callosal dysplasia. Fetal movement poses a great challenge in the collection of corpus callosum information; however, visualization in the mid-sagittal plane of the brain is possible by ultra-fast localization using the axial and coronal planes [51]. In light of the high prevalence of complex forms of CCA over isolated forms, fetal MRI is advantageous and provides more information about cortical morphology, especially when performed after 27 weeks of gestation. Ultrasound may not adequately detect malformations in cortical development such as gray matter heterotopia, polymicrogyria, and anomalies of sulcation such as lissencephaly, schizencephaly, and pachygyria. Moreover, in nearly 83% of patients, fetal MRI-identified abnormalities were not revealed by antenatal ultrasonography [43]. In addition, fetal MRI can suggest a possible etiology for CCA, such as developmental malformations that can give a clue for a genetic syndrome, and disruptive changes in the fetal brain parenchyma may suggest an acquired or metabolic abnormality associated with CCA [51]. Furthermore, by utilizing fetal MRI, we can detect subtle other associated CNS abnormalities that may indicate postnatal neurodevelopmental outcomes [52]. Fetal MRI has better contrast resolution, multiplanar capability, and allows direct visualization of the corpus callosum [26]. A simple system-based classification based on a fetal MRI midline sagittal image is the first step in the evaluation of CCA; however, future studies may use advanced imaging techniques like functional MRI or fractional anisotropy for precise anatomic subclassification [20]. The use of machine learning models in conjunction with genetic and clinical data to analyze prenatal brain multi-parametric MRI data may improve diagnosis and postnatal outcome monitoring [53]. The deep learning-facilitated pipeline helps radiologists select good-quality fetal brain images in a shorter time frame and facilitates anatomical measurements [54]. Multidisciplinary collaborative teams are involved, including genetics, obstetrics, perinatology, pediatric surgery, neurosurgery, and neurology, in the management of CCA [55].

Our study used Hanna et al. classification to describe heterogeneity in CCA morphology that could describe our entire cohort [20]. Recent studies of fetal MRI have used this classification, including a study performed by El Ameen et al. in 2019 and Manor et al. in 2020 [26,56]. The study performed by El Ameen et al. used this classification to evaluate the role of DTI and fiber tractography in children with CCA and also studied the clinico-radiological correlation [56]. Manor et al. described fetal MRI in CCA in their pictorial essay based on this classification [26]. The previous classification systems based on prenatal ultrasonography divided CCA broadly into complete and partial agenesis, and thus did not account for heterogeneity in callosal morphologies. The extensive spectrum of morphological descriptions of CCA may shed light on the time of insult, etiology, and prognostic implications.

In the present study, the classification of CCA was conducted into three categories: complete, hypoplasia, dysplasia, and hypoplasia with dysplasia. The largest subgroup consisted of individuals with complete agenesis (79.1%), which is similar to the findings reported by Ruland et al., in which they reported the majority percentage of complete agenesis cases (76%) [57]. These findings are in line with a study conducted by Ibrahim et al., in which they diagnosed 77.7% of cases of complete agenesis [44]. Correspondingly, Al-Hashim et al., reported in their study the largest subset of patients with complete agenesis (52%) [19]. The complete agenesis group exhibited a higher prevalence of ventriculomegaly, colpocephaly, and Probst bundles. Ventriculomegaly was not considered an additional finding, as can be seen with corpus callosal agenesis. In our study, colpocephaly (58.3%) was more frequent in cases of complete agenesis than in hypoplasia or dysplasia, similar to the study performed by Neal et al. [21]. Colpocephaly occurs due to a decrease in white matter in the occipital cortex, resulting in the subsequent enlargement of the posterior horns of the lateral ventricles [1,58]. On the other hand, in the group with hypoplasia or dysplasia, the presence of intact myelinated callosal tracts is likely responsible for preserving the structural integrity, which in turn prevents the enlargement of the posterior horns of the lateral ventricles to a certain extent. Probst bundles were observed in cases of complete agenesis of the corpus callosum but were not found in cases of hypoplasia, consistent with the findings reported by Hett et al. [1]. Furthermore, Probst bundles are found in 3/4 (75%) cases of isolated CCA and not seen in complex forms of CCA, similar to findings reported by Byrd et al. [23]. However, the identification of Probst bundles on real-time fetal MRI can be challenging, possibly explaining the lower percentage of Probst bundles in the study cohort compared to post-natal research studies. Cavum septum pellucidum is considered present, even if incomplete or present in the form of remnants. This fact probably explains the smaller number of absent CSP cases in our cohort. Most cases of CSP are identified as cases of complete agenesis. These findings are in contrast with a study performed by Neal et al., who reported that septal anomalies are most common in the hypoplasia class [21].

The complex form of CCA can be further classified as associated with neurological, non-neurological, or both. Out of 24 patients in our analysis, only 4 (16%) had isolated CCA; the remaining 20 (83.3%) cases were linked to other abnormalities. This is consistent with the study performed by Ruland et al., which found that 71% of cases were non-isolated forms of CCA and 29% of cases were isolated forms. [57]. Our study results are also in line with similar work reported by Ibrahim et al., in which they revealed that 18.5% had isolated CCA and 74% were in non-isolated form [44]. Manganaro et al. conducted a study in which they discovered that fetal MRI diagnosed 26.9% isolated CCA, with the majority of the cohort (about 73.1%) being associated with other abnormalities [42]. This maximized prevalence of non-isolated CCA over isolated CCA is an important predictor of postnatal prognosis, according to recent data on autopsy studies [59,60,61]. 

Associated CNS anomalies were seen in 14/20 cases (70%), which is in line with studies conducted by Al-Hashim et al. [19]. Interhemispheric cysts were the most frequent related neurological findings in around 9/20 cases (45%), which correlates well with the study performed by Ibrahim et al., who reported that IHC is the most common neurological finding, representing up to 37.5% [44]. In another study performed by Manganaro et al., it was found that around 19.7% of cases had IHC [42]. Additionally, these findings are similar to the study performed by Byrd et al., with 35/79 cases (44%) having associated IHC [23]. However, in another study performed by Hetts et al., only 14% of cases of corpus callosal agenesis had IHC [1]. Our studies had (20%) cases of migrational anomalies, which is in accordance with the study by Hetts et al. (25%), Byrd et al. (23%), and others [1,23]. In our study between CCA classes, migrational anomalies were more common in the complete agenesis group in comparison with the hypoplasia group, similar to Hetts et al., who reported migrational anomalies in 29% of cases with agenesis of corpus callosum and 21% of cases with hypogenesis [1]. 

In our study, vermian hypoplasia represented up to 10% of posterior fossa anomalies, which was less when compared to research by Manganaro et al. and Ibrahim et al. that found up to 30% of related cerebellar malformations [42,44]. Furthermore, Byrd et al. reported that 7% of cases had vermian hypoplasia [23]. 5% of our cases had the Dandy Walker variant with an abnormal and distorted medulla, which is lower than the study performed by Neal et al. and Byrd et al. [21,23]. Neural tube defects are noted in 5% of cases, which is in line with the study performed by Byrd et al. [23] and lower when compared to the study performed by Ibrahim et al. [44].

6/24 cases (25%) of the total cohort had associated non-CNS anomalies, which is similar to the study performed by Taylor et al. [62]. The associated only non-neurological anomalies were found in two cases, in which one case had T12 hemivertebra with thoracic vertebral body fusion but was suspected to have Aicardi syndrome, and another case had a nasal dermoid cyst, which is a rare syndrome known as fronto-nasal dermoid cysts and agenesis of corpus callosum [63]. 

The cranio-facial abnormalities were the most common non-neurological anomalies detected in the study. This is according to the studies performed by Jeret et al. and Bedeshi et al. [12,36]. 4 cases (20%) showed combined neurological and non-neurological anomalies, which is higher than the study performed by Ibrahim et al. (10%) [44]. Of these, the first case had facial dysmorphism, including frontal bossing secondary to severe hydrocephalus with associated fused eyes. The second case includes a non-neurological anomaly of the cleft lip with a cleft palate and an associated sphenoidal meningomyelocele as a neurological anomaly. The neurological anomalies associated with this case include hemimegalencephaly and germinolytic cysts. Another case included non-neurological anomalies diagnosed as hydroureteronephrosis, which showed a neurological anomaly of heterotopia. The last case had trisomy 8 and an associated thick nuchal fold with an interhemispheric cyst. The study performed by Ibrahim et al. reported 2 cases of combined anomalies; one case had bilaterally enlarged cystic kidneys and posterior fossa cysts. Another case had a bilateral clubfoot with a large interhemispheric cyst and posterior fossa anomalies [44].

This study highlights the importance of diagnosing CCA abnormalities with fetal MRI, which has implications for the genetic counselling of parents about future pregnancy risks. Consequently, greater research on the significance of fetal MRI is needed, as behavioral and cognitive issues may not be recognized until school age, as corpus callosal abnormalities have been associated with autism and schizophrenia. Future studies using diffusion-weighted and diffusion tensor imaging will provide more insight into white matter abnormalities in fetuses with CCA. It is possible to detect abnormal brain development earlier by using diffusion-weighted imaging (DWI) methods, which are sensitive enough to detect microstructural changes in cerebral tissues. Moreover, ADC values help evaluate the etiology and evolution of various neurological abnormalities in fetuses. The rapid acquisition protocol and artifact correction made it possible to detect fetal brain development anomalies within a short timeframe [64].

The study Is limited by a retrospective study design. The relationship between various CNS and non-CNS abnormalities and different CCA subtypes could not be analyzed owing to the small sample size. The study included only fetuses diagnosed with CCA by prenatal ultrasound. Fetal ultrasonography is operator-dependent, and diagnosis depends on the obstetrician’s judgment. Fetal MR imaging conducted at later gestational ages has better sensitivity, though it is clinically impractical. It is therefore important to perform fetal MRI at a consistent gestational age to maximize the study’s accuracy. Replications of the study at different institutions may yield different results. As this study was conducted at a tertiary health care referral center, a higher proportion of fetuses are seen with associated anomalies, which makes our results less generalizable.

## 5. Conclusions

The prevalence of complex forms of CCA is higher than that of isolated CCA. Fetal MRI is recommended in the characterization of CCA, as in many instances of apparent isolated CCA. It is the main method to differentiate between isolated CCAs and complex CCAs. It provides additional information about the type and severity of CCA, detects associated anomalies, and helps to comprehend the underlying mechanisms. The study emphasizes that, in many instances, a single midline sagittal MRI image is adequate for identifying an abnormal corpus callosum. However, in cases where the conclusion is unclear, it is advisable to examine the fetal CC in all three planes to reduce interpretation errors. The inclusion of the CCA subclasses’ classification accurately describes the heterogeneity of callosal morphology and its associated prognosis. It is recommended that the classification scheme include the nomenclature of Probst’s bundles, which is commonly seen in sporadic cases of complete callosal agenesis and has significance in the post-natal outcome. We recommend that it is best practice standard to include clinical genetics review, and CMA testing in patients with both isolated and non-isolated forms of CCA. The integration of morphological classification combined with clinical and genetic information using deep machine learning techniques allows for further understanding of an individual with CCA. Future studies should use functional and tractography-based imaging techniques for early detection of white matter abnormalities, Probst tracts, and other faulty fibers in fetuses with CCA.

## Figures and Tables

**Figure 1 diagnostics-14-00430-f001:**
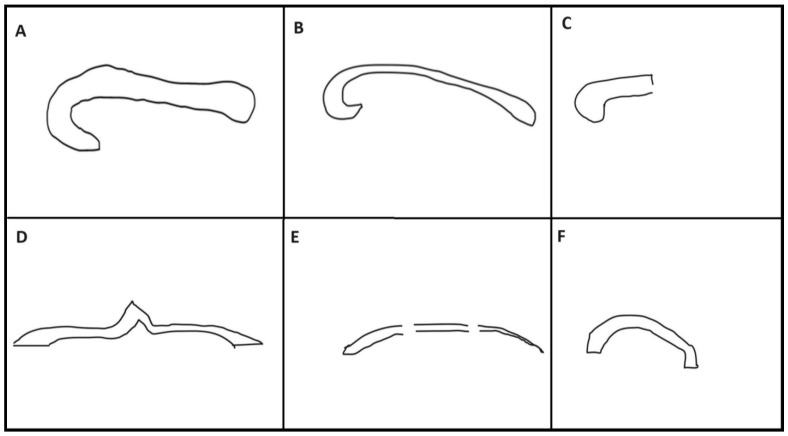
Schematic representative diagram showing structural morphology of normal corpus callosum and corpus callosal abnormalities on the midline sagittal fetal MRI. (**A**) Normal corpus callosum showing rostrum, genus, body, and splenium. Corpus callosal abnormalities are divided into subclasses: (**B**) Hypoplasia without dysplasia—uniform thinning with the intact structure of corpus callosum having all four parts. (**C**) Hypoplasia of posterior region with anterior remnant- Absent posterior corpus callosum. (**D**) Dysplasia without hypoplasia- this represents a structurally abnormal corpus callosum with no thinning. (**E**) Hypoplasia with dysplasia—striped type- lack of the structurally distinct genus and splenium with uniformly thin stripe corpus callosum. (**F**) Hypoplasia with dysplasia—kinked type- hypoplasia and kinking of anterior or posterior aspect. Complete, agenesis—Absence of corpus callosum at fetal MRI.

**Figure 2 diagnostics-14-00430-f002:**
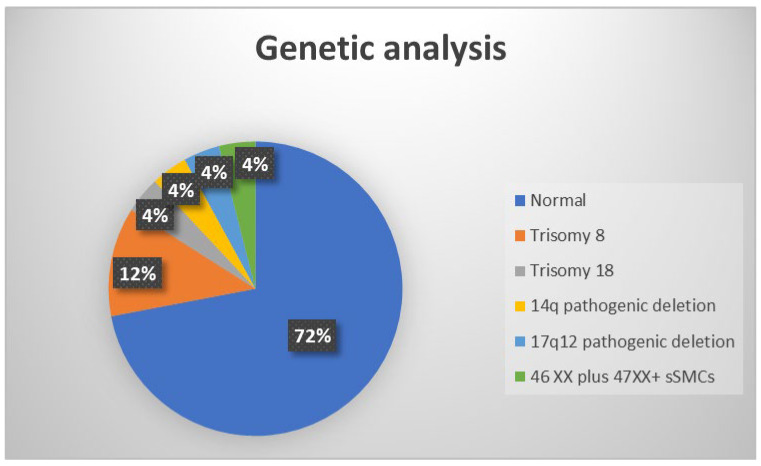
Illustrative chart showing genetic analysis results of the study population. sSMCs, small supernumerary marker chromosomes.

**Figure 3 diagnostics-14-00430-f003:**
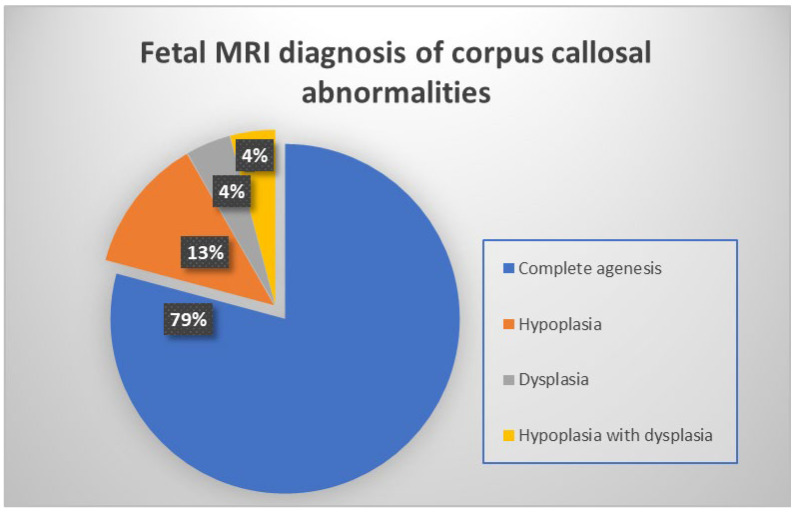
Illustrative chart showing frequencies of various corpus callosal abnormalities detected on fetal MRI.

**Figure 4 diagnostics-14-00430-f004:**
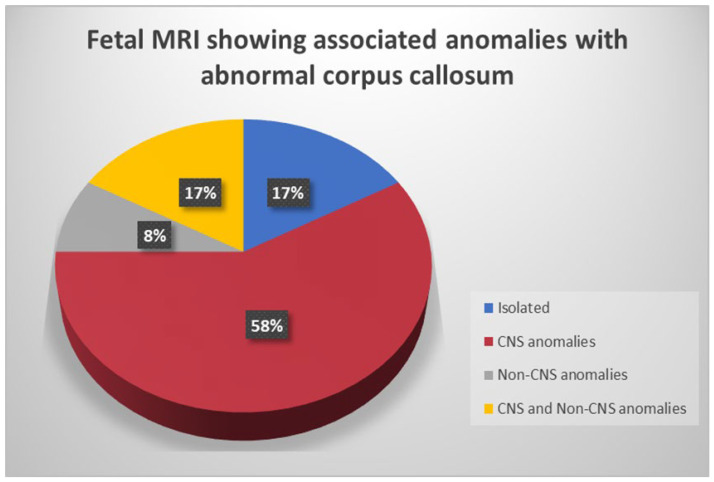
Fetal MRI showing various anomalies associated with abnormal corpus callosum.

**Figure 5 diagnostics-14-00430-f005:**
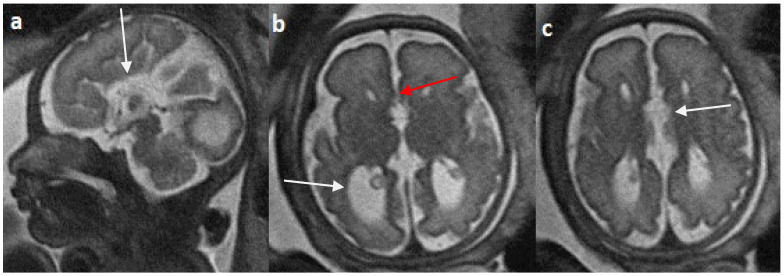
Fetal MRI images of a 31 gestational weeks aged fetus with complete agenesis of the corpus callosum. (**a**) A T2-weighted sequence image in sagittal plane shows that the corpus callosum is not visible (white arrow). (**b**) A T2-weighted sequence image in axial plane shows a “teardrop”-like dilation in the posterior horn of bilateral lateral ventricles suggestive of colpocephaly (white arrow) and absence of corpus callosum and septum pellucidum (red arrow) in the midline. (**c**) A T2-weighted axial image shows a small interhemispheric fluid collection/cyst in the midline. (arrow). No Probst bundles are observed.

**Figure 6 diagnostics-14-00430-f006:**
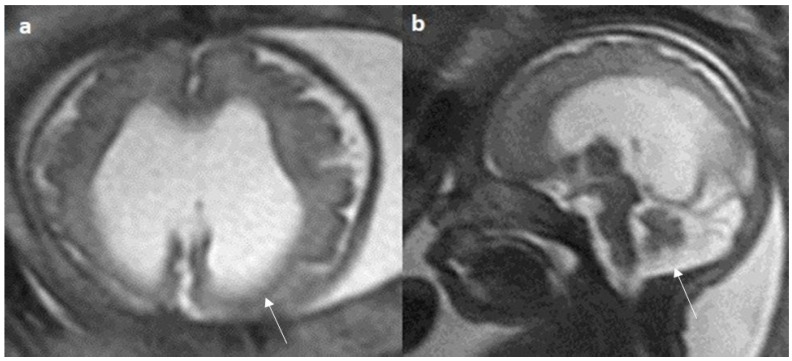
Fetal MRI images of a 29 gestational weeks aged fetus in a case of hypoplasia with dysplasia of the corpus callosum. (**a**) A T2-weighted sequence in axial plane showing severe dilatation of the lateral ventricles, particularly at the body and atria, with thinning of the brain parenchyma in the posterior parieto-occipital region, and an associated absent cavum septum pellucidum was noted (white arrow). (**b**) A T2-weighted sequence in midsagittal plane shows thin anterior corpus callosum with dysplasia. Additionally, hypoplasia of the inferior vermis is noted (white arrow).

**Figure 7 diagnostics-14-00430-f007:**
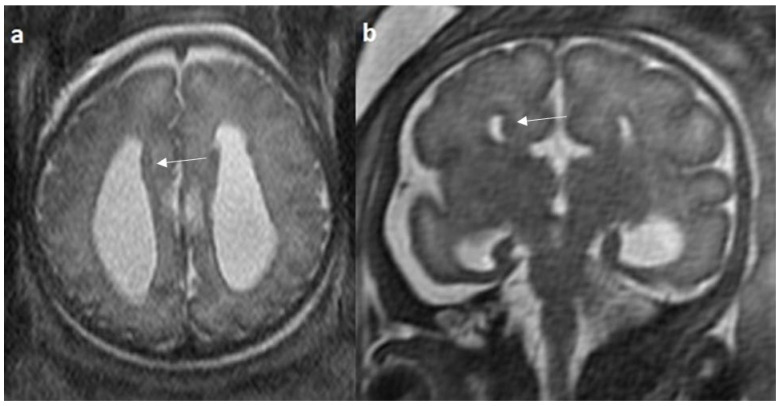
Fetal MRI images of a 20 gestational weeks aged fetus with complete agenesis of the corpus callosum. (**a**) A T2-weighted sequence image in axial plane showing non-decussating anterior-posterior white matter tracts known as Probst bundles medial to the lateral ventricles (arrow). (**b**) A T2- weighted sequence coronal image showing Probst bundles indenting superomedial margins of lateral ventricles. Probst bundles are seen with complete agenesis of the corpus callosum.

**Figure 8 diagnostics-14-00430-f008:**
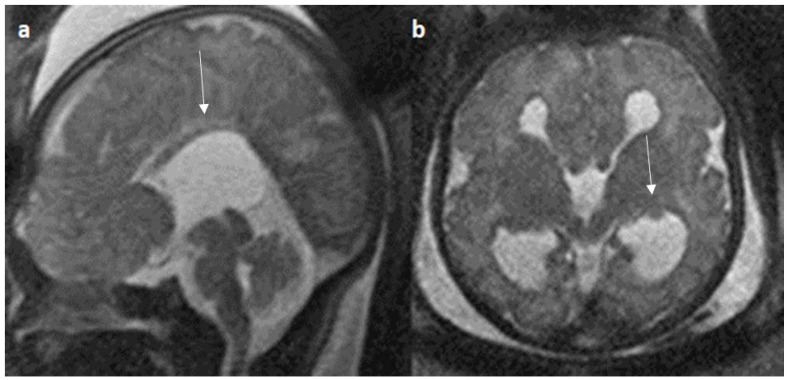
Fetal MRI images of a 34 gestational weeks aged fetus, with complete agenesis of the corpus callosum and gray matter heterotopia. (**a**) A T2-weighted sequence image in mid sagittal plane showing complete absence of the corpus callosum with secondary changes in ventriculomegaly. (**b**) A T2-weighted axial sequence image showing associated subependymal heterotopia (white arrow) along the occipital horns of the bilateral lateral ventricles with moderate dilatation of lateral and third ventricles.

**Figure 9 diagnostics-14-00430-f009:**
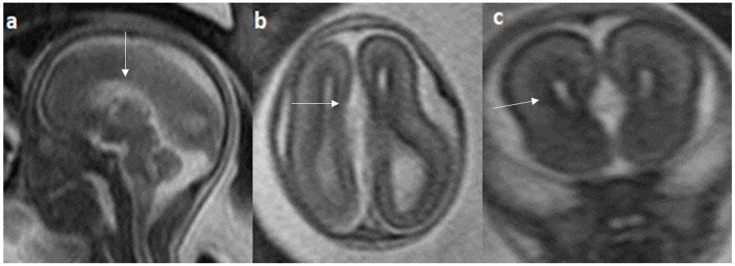
Fetal MRI images of a 20 gestational weeks aged fetus with complete agenesis of the corpus callosum. (**a**) A T2-weighted sequence image in axial plane shows non-visualization of the corpus callosum, suggestive of complete agenesis. (**b**) A T2 weighted sequence axial image showing a well-defined extra axial fluid in the interhemispheric fissure. A lack of septum pellucidum was also observed. (**c**) The coronal plane shows an “steer horn” sign (arrow) in the anterior horn of the bilateral lateral ventricles and the absence of the corpus callosum and septum pellucidum in the midline region.

**Figure 10 diagnostics-14-00430-f010:**
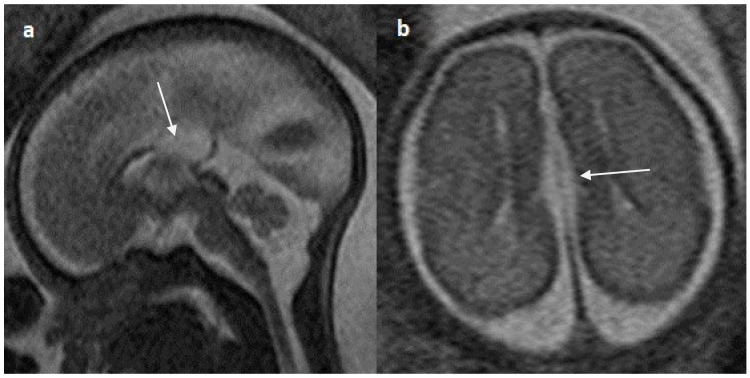
Fetal MRI images of a 26 gestational weeks aged fetus with hypoplasia of the corpus callosum. (**a**) A T2-weighted sequence in mid sagittal plane image shows normal anterior part of corpus callosum with absent posterior part (arrow), suggesting hypoplasia of posterior region with anterior remnant, rest of the brain parenchyma in this section is normal. (**b**) A T2-weighted sequence in axial plane showing a small midline interhemispheric fluid collection/cyst without communication with the lateral ventricles.

**Table 1 diagnostics-14-00430-t001:** Maternal and birth characteristics of the study population.

Fetuses with Corpus Callosal Abnormalities (N = 24)	Mean ± SD or N (%)
Mean maternal age at delivery (years)	30 ± 7
Mean WG (weeks) at the time of fetal MRI	26 ± 5
Maternal comorbid diagnoses	
Preeclampsia	5 (21%)
Gestational or chronic hypertension	3 (13%)
Gestational diabetes	3 (13%)
Maternal cancer	1 (4%)
Teen pregnancy	4 (17%)
Elderly pregnancy	5 (21%)
Amniocentesis	
Normal karyotype, CMA and WES	17 (71%)
Abnormal results	7 (29%)
Birth Weight (g)	2947 ± 951
Birth WG (week)	36 ± 5
Male	13 (55%)
Disposition	
Inborn live birth	20 (83%)
MTP	2 (8%)
Stillbirths	2 (8%)

SD, standard deviation; N, number; MRI, magnetic resonance imaging; WG, weeks of gestation; g, grams; NICU, neonatal intensive care unit; MTP, medical termination of pregnancy; CMA, chromosomal microarray analyses; WES, whole exome sequencing.

**Table 2 diagnostics-14-00430-t002:** Fetal MRI characteristics of 24 patients with corpus callosal abnormalities, including neurological, non-neurological anomalies, and genetic analysis.

No.	WG/Sex	Delivery History	CCA Subclass	Associated CNS Anomaly	Associated Non-CNS Anomaly	Genetic Analysis
1	36/M	Full term	Complete Agenesis	colpocephaly		46XY
2	34/F	Full term	Complete Agenesis	ventriculomegaly, heterotopia		46XX
3	24/M	Full term	Complete Agenesis	IHC, Arachnoid cyst, colpocephaly, ventriculomegaly	Thick nuchal fold, hydronephrosis	Trisomy 8
4	22/F	Full term	Complete Agenesis	colpocephaly, ventriculomegaly		46XX
5	20/M	Preterm	Complete Agenesis	IHC, absent septum pellucidum, blake pouch cyst, vermian hypoplasia		Trisomy 18
6	23/M	Preterm	Complete Agenesis	IHC, colpocephaly		46XY
7	31/F	N/A	Complete Agenesis	IHC, colpocephaly		46XY
8	22/F	Full term	Complete Agenesis	Heterotopia		46XX
9	25/M	Full term	Complete Agenesis	IHC, colpocephaly		8q partial trisomy syndrome
10	20/M	Full term	Complete Agenesis	IHC		46XY
11	24/M	Full term	Complete Agenesis	Large IHC with incorporation of left ventricle into cyst		46XY
12	19/F	N/A	Complete Agenesis	Dandy walker variant with distorted medulla, cerebellar hypoplasia		46XX, 47XX + sSMCs
13	27/F	Full term	Complete Agenesis	Sphenoidal meningoencephalocele	Cleft lip, cleft palate	46XX del 17q12
14	25/M	Full term	Complete Agenesis	Colpocephaly	Nasal dermoid cyst	46XY
15	29/M (twin)	Full term	Complete Agenesis	IHC, colpocephaly		46XY
16	26/M	N/A	Complete Agenesis	IHC, colpocephaly, ventriculomegaly		46XY
17	33/M	Full term	Complete Agenesis	Colpocephaly, Absent septum pellucidum, ventriculomegaly	T12 hemivertebrae, thoracic bodies fusion	46XY
18	27/F	Full term	Complete Agenesis	Colpocephaly, Absent septum pellucidum, ventriculomegaly		46XX
19	28/M	Preterm	Complete Agenesis	Colpocephaly, Absent septum pellucidum, heterotopia	Renal anomaly (hydroureteronephrosis)	Trisomy 8 mosaicism
20	27/F	Full term	Hypoplasia of splenium	Colpocephaly		46XX
21	26/M	N/A	Hypoplasia of splenium	IHC, blake pouch cyst		46XY
22	29/F	Preterm	Complete Hypoplasia	Absent septum pellucidum, ventriculomegaly, vermian hypoplasia, aqueductal stenosis		46XX del 14q24.3 and q32.1.
23	22/F	Preterm	Dysplasia	Hemimegalencephaly, germinolytic cyst, schizencephaly	Facial dysmorphism with fused eyes	46XX
24	26/F	Preterm	Hypoplasia with Dysplasia	Ventriculomegaly, Reduced white matter		46XX

CNS, central nervous system; N/A, not available; CCA, corpus callosal abnormalities; WG, weeks of gestation; IHC, interhemispheric cyst; sSMCs, small supernumerary marker chromosomes.

## Data Availability

The data presented in this study are available on request from the corresponding author (Data available on request due to privacy and ethical reasons).

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
