# Peer review of "Fetal MRI Analysis of Corpus Callosal Abnormalities: Classification, and Associated Anomalies"

_diagnostics, 2024, doi:10.3390/diagnostics14040430_

Round 1

Reviewer 1 Report

Comments and Suggestions for Authors

In a retrospective study regarding anomalies of the corpus callosum (CCA) in 24 fetuses, the authors report results obtained by evaluating the corpus callosum morphology from T2-weighted MRI images.

I have some suggestions for making the manuscript interesting to readers.

-       Authors should report the linear resolution of the MRI images and comment on the reported results in light of the resolution of the images. As an example, compare CC size to voxel size.

-       The authors should explain their statement regarding: “future studies using diffusion-weighted and diffusion tensor imaging will provide more information about white matter abnormalities in fetuses with CCA” (p. 15 lines 33-35). Why will diffusion studies provide more? In particular, what aspect will they highlight? For this reason, the authors can read the following paper: MG Di Trani, L Manganaro, A Antonelli, M Guerreri, R De Feo, C Catalano, S Capuani, Apparent diffusion coefficient assessment of brain development in normal fetuses and ventriculomegaly, Frontiers in Physics 2019;7:160. https://doi.org/10.3389/fphy.2019.00160

-       It would also be useful for readers to know the incidence rate of ventriculomegaly in the CCA cases analyzed.

Author Response

Thank you very much for taking the time to review this manuscript. Please find the detailed responses in the below attachment and the corresponding revisions in track changes in the re-submitted files.

Reviewer 2 Report

Comments and Suggestions for Authors

Esteemed Author Team,

I read with interest your Manuscript ID: diagnostics-2838735 which is is an interesting, well-written, original article describing the fetal MRI analysis of corpus callosum abnormalities. In the article you have shown Hanna et al. classification of CC pathologies and you’re also reviewing some associated CNS and non-CNS anomalies.

While such analysis  has been done before, however few studies are based on that latest classification system. None of the mentioned classification systems is widely used and no consensus is present regarding the heterogeneous terminology of callosal abnormalities

Few suggestions and remarques are listed below:

  1. The Conclusion part of the abstract is with bigger font size comparing to the rest of the text.
  2. You are describing in details the various classifications. Do you consider Hanna’s classification more user-friendly comparing to the others? How the role of genetic mechanisms in CCA could be implemented to a morphological classification system? Are you sure reference 21? is correct?
  3. In METHODS you mention that “…suspected related congenital anomalies were not adequately assessed on ultrasound”. Would you please, precise the concrete anomalies that are not adequately seen on US.
  4. In RESULTS you саn add that you routinely perform at your institution genetic analysis in fetuses with ACC.  Did you also observed a higher frequency  of chromosomal malformations in your teen or elderly pregnancies groups?
  5. In FIGURE 5c “a small interhemispheric cyst in the midline” is not really  evident? The arrows are not at the precise location in some of the other figures.
  6. I agree completely with the statement in the DISCUSSION that 3D reconstructions or volumetric studies are not necessary as corpus callosum is well seen on sagittal view , however would you consider  volumetric segmentation useful especially in cases of callosal dysplasia?

General comments: English language is excellent. The methods are adequately described and  the results are clearly presented. Please, check again the instructions of the journal for the references and correct them. In some of them there are “.” and “;” following the first and last name of the authors, should be standardized/unified on the correct way.

Overall, I find the data valuable and only minor corrections are needed before publication. The Introduction could be shorter. The Discussion is very well written and the  Results section is explained in the text and illustrated by table 1 and 2.

My overall impression of the manuscript is excellent

You саn make visible the subtle changes as Track Changes On.

Thank you!

Author Response

(The authors gave the same response as above.)

Reviewer 3 Report

Comments and Suggestions for Authors

This is a descriptive study on a cohort of 24 fetuses with corpus callosum abnormalities (CCA).

The study seeks to emphasize the diagnostic role of prenatal MRI in characterizing heterogeneous corpus callosal disorders using the classification system according to Hanna RM, et.al. Neurology, 2011.

Retrospective data from pregnant women who underwent fetal MRI were analyzed for callosal anomalies. The fetal maternal unit sent all of these cases to the diagnostic neuroradiology department for fetal MRI after discovering callosal abnormalities on prenatal ultrasound. Patients were also evaluated for related neurological and non-neurological abnormalities.

The authors reported complete callosal agenesis in 79.1%, followed by hypoplasia (12.5%), dysplasia (4.2%), and hypoplasia with dysplasia (4.2%). Only 17% of patients had isolated CCA, whereas the vast majority (83%) had complicated forms of CCA associated with other CNS and non-CNS abnormalities. Of the non-isolated CCA, 58% were associated with other CNS anomalies, 8% with non-CNS anomalies, and 17% contained both.

The author concluded that fetal MRI is useful for classifying corpus callosum abnormalities. This methodology is quite useful for discriminating between isolated and complicated types of CCA.

This is a well-written manuscript, although several topics need to be addressed in this study:

1.      Hanna RM, et.al. (Neurology. 2011), found that their classification of CCA is helpful for prognostic implications on the postnatal outcome. Therefore, the clinical outcome in terms of neurological and developmental examination in the sub-classes of fetal CCA should be addressed.

2.      Did the pregnancies with CCA undergo whole exome sequencing (WES)? If WES was performed, the results should be reported.

3.      In Figure 5. “Fetal MRI images of a 31 gestational weeks aged fetus with complete agenesis of the corpus callosum. a) A T2-weighted sequence image in sagittal plane shows that the corpus callosum is not visible (red arrow)...”. There is no red arrow in this panel.

4.      Figure 6. “Fetal MRI images of a 29 gestational weeks aged fetus in a case of hypoplasia with dysplasia of the corpus callosum a) A T2-weighted sequence in axial plane showing severe dilatation of the lateral ventricles, particularly at the body and atria, with thinning of the brain parenchyma in the posterior parieto-occipital region An associated absent cavum septum pellucidum was noted (white arrow).b) A T2-weighted sequence in mid sagittal plane shows thin anterior corpus callosum with dysplasia. Additionally, hypoplasia of the inferior vermis is noted (white arrow).” All arrows are in the wrong position.

Author Response

(The authors gave the same response as above.)
